# Diagnosis of Systemic Rheumatic Disease Using the Connective Tissue Disease Screen

**DOI:** 10.3390/antib14030056

**Published:** 2025-07-02

**Authors:** Abeline Kapuczinski, Dorian Parisis, Nour Kassab, Julie Smet, Muhammad Soyfoo

**Affiliations:** 1Department of Rheumatology, Hôpital Erasme, H.U.B, Université Libre de Bruxelles, 1070 Brussels, Belgium; abeline.kapuczinski@ulb.be (A.K.); dorian.parisis@ulb.be (D.P.);; 2Laboratory of Immunology, Laboratoire Hospitalier Universitaire de Bruxelles—Université Libre de Bruxelles (LHUB-ULB), 1000 Bruxelles, Belgium; julie.smet@chu-brugmman.be

**Keywords:** autoimmune diseases, autoantibodies, diagnostic performance, screening assays

## Abstract

Connective tissue diseases (CTDs) comprise a heterogeneous group of autoimmune conditions characterized by diverse clinical manifestations and autoantibody profiles, posing significant diagnostic challenges. This systematic review and meta-analysis evaluated the diagnostic performance of automated connective tissue disease screening assays, commonly known as CTD screens, in diagnosing systemic rheumatic diseases. Eleven studies, including cohort and case–control designs, involving a total of 2384 CTD-positive patients, 8972 controls without CTD, and 679 healthy blood donors, were analyzed. The results demonstrated a pooled sensitivity of 79.36% and specificity of 90.79% for Elia^®^ CTD-screen, and a sensitivity of 87.23% and specificity of 83.56% for QuantaFlash^®^ CTD-screen. These tests exhibited varied sensitivity across individual CTDs, with excellent specificity for distinguishing CTD patients from healthy controls. Despite their utility, CTD screens should not be solely relied upon for diagnosis due to limitations in positive predictive value, particularly in low-prevalence populations. Clinical context and expert rheumatological evaluation remain indispensable. Optimizing the use of CTD screens can enhance diagnostic efficiency, reduce unnecessary testing, and mitigate patient anxiety and healthcare costs. Further research focusing on integrating these assays with clinical evaluation is recommended.

## 1. Introduction

Connective tissue diseases (CTDs) encompass a heterogeneous group of rheumatic diseases with a broad spectrum of clinical and biological characteristics characterized by organ involvement [1,2]. As such, specific and characteristic clinical signs and symptoms as well as autoantibodies define specific connective tissue diseases [3]. There are mainly five defined autoimmune connective tissue diseases: systemic lupus erythematosus (SLE), systemic sclerosis (SSC), myositis, rheumatoid arthritis (RA), and Sjogren’s syndrome (SS). One of the principal conundrums in defining and classifying CTD lies in the fact that some patients portray features of autoimmune disease but do not satisfy classification criteria for a defined CTD. These patients are diagnosed as undifferentiated connective tissue disease (UCTD). In addition, other patients have criteria for two or more defined autoimmune conditions and are diagnosed as having an “overlap syndrome”, of which mixed connective tissue disease (MCTD) is part of it [3].

Due to their heterogeneity and complexity, the diagnosis of CTD is often challenging [2]. Autoantibodies are a helpful tool in enabling the diagnosis of CTD as some autoantibodies are clearly associated with a specific phenotype of CTD [1]. Antinuclear antibodies (ANAs) are valuable laboratory markers for screening systemic rheumatic diseases [4].

However, autoantibodies carry certain limitations, making their interpretation sometimes difficult. In particular, autoantibodies can be associated with more than one disease [1]. Moreover, ANAs can also be found in patients with several broad diseases and even in healthy individuals [5]. Although ANA positivity shows high sensitivity for several systemic rheumatic diseases, their presence is non-specific and may result from environmental exposures, malignancies, drugs, or infections [6].

In current clinical practice, there are several lines of evidence corroborating the inappropriate use of ANA screening, thereby leading to unnecessary visits and economic costs, as well as patient anxiety. Understanding how to use ANAs is cardinal to reduce unnecessary referrals and costly workups [6]. ANA screening should be avoided in patients with low pretest probabilities for ANA-associated rheumatic diseases [7]. Nevertheless, early in the disease, patients with systemic rheumatic disease often present with vague symptoms such as fatigue, joint pain, or muscle weakness, with a wide array of possible diagnoses [8]. Whether ANAs are useful biomarkers in this particular clinical setting, in particular for the identification of early disease and management, is still debated [9].

ANAs remain useful for ANA-associated systemic rheumatic disease diagnosis. Indirect immunofluorescence (IIF) assay on cultured human epithelial carcinoma cells (Hep-2 cell) has been used as a gold standard method [10]. However, it has been shown that some subtypes of ANA, especially anti-SSA/Ro and anti-JO 1 antibodies, may be overlooked by IIF. Furthermore, IIF requires experienced and well-trained analysts, is time-consuming, and shows high inter-observational variability [9]. Recently, commercially available automated CTD-associated ANA screening assays, the so-called “CTD screen”, have been developed, allowing the simultaneous detection of several antibodies [8]. CTD screen has a higher specificity but a lower sensitivity than IIF [8]. CTD screen was shown to be excellent for patients with SLE, and the combination with IIF was more effective for the diagnosis of systemic rheumatic diseases [5,10].

The use and interpretation of the CTD screening test according to cut-offs (“negative”, “doubtful”, “positive”) is very challenging for the clinician. The aim of this study was to analyze the use and contribution of searching ANAs and CTD screen to diagnose systemic rheumatic disease with a systematic review of the literature with a meta-analysis.

## 2. Materials and Methods

A systematic review of the literature was carried out according to the Cochrane Collaboration for a systematic review of diagnostic tests studies. Automated research was conducted with different MeSH and free text so that an equation was obtained for each database: PubMed, Scopus, Ovid, and Cochrane Library databases (from inception to 24 February 2021), without either language or publication period restrictions.
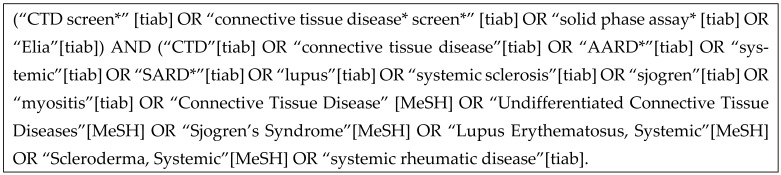


The search was completed with manual search of used references in firm documentations or journals including the CTD screen. All cohorts or case–control studies having considered CTD screen as diagnostic tool for connective tissue disease were imported in a spreadsheet software program (Excel) using a bibliographic management software (Zotero. Version4) to be selected according to title and abstract by two independent investigators after removing the duplicates. The two investigators compared their results to discuss discrepancies, and a third party was designated in case of a disagreement. Selected studies were evaluated on full text for eligibility for inclusion in the systematic review of the literature. Moreover, studies had to specify the CTD screen positivity rate to allow the meta-analysis of data.

Figure 1 shows the flowchart of the selection of the different references used in this meta-analysis. Inclusion criteria were all case–control or cohort studies assessing CTD screen to diagnose connective tissue disease. A total of 685 potentially eligible references were identified according to our different equations and research strategy. After the exclusion of 392 duplicated articles, titles and abstracts from the 251 remaining articles were analyzed. From this, 42 articles were analyzed from the full text and 31 references were excluded because they did not satisfy our criteria (not case–control or cohort studies). Finally, 11 articles were included for more analyses. This study was conducted in accordance with the Preferred Reporting Items for Systematic Reviews and Meta-Analyses (PRISMA) guidelines.This systematic review was conducted according to PRISMA guidelines.

### Statistical Analysis

CTD screen positivity rates were systematically extracted for sub-groups of interest provided in the studies. When unavailable, positivity rates were calculated from other available data. Sensitivity, specificity, and their 95% confidence intervals were calculated for each type of study (cohort or case–control) and each commercial kit. Quality of studies was evaluated according the QUADAS-2 score tool (Quality Assessment of Diagnostic Accuracy Studies) [11]. Because of the study design bias, the random-effects model was used to obtain pooled sensitivity and specificity and their confidence interval. Results were presented as Forest’s diagrams. In each Forest plot, symbol size corresponds with sample size. To evaluate the heterogeneity of included studies in the meta-analysis, we used the I^2^. A value of 0 to 25% of the I^2^ value represents non-significant heterogeneity, 26 to 50% represents poor heterogeneity, 51 to 75% represents moderate values of heterogeneity, and >75% represents high heterogeneity. Publication bias was evaluated with visual inspection of the funnel plot and Egger and Begg’s tests [12,13]. Statistical analysis was performed with MedCalc (MedCalc Softwares, Belgium). Statistical significance was considered significant for a *p*-value < 0.05.

## 3. Results

From the 11 articles selected for meta-analysis, 15 datasets were analyzed according to the type of recruitment (8 cohort studies and 7 case–control studies) or the kit used (9 Elia^®^ CTD-screen, 5 QuantaFlash^®^ CTD-screen, 1 Varelisa^®^ CTD-screen, and 1 Elia^®^ and/or QuantaFlash^®^ CTD-screen). The 11 studies encompassed 2384 patients diagnosed with CTD, 8972 patients without CTD, and 679 blood donors (Table 1).

Quality assessment of the studies according the QUADAS-2 score tool was performed (Table 2). Overall, quality assessment indicated that studies were moderate quality, in particular for patients’ selection. Indeed, several studies carried out case–control studies and resorted to patients from blood banks. Moreover, in some studies, final diagnosis was confirmed only in IIF patients and/or CTD-screen positive and in part of CTD-screen negative [8,14,15]. Jeong et al. did not clearly explain their patients’ selection [5,10].

Pooled sensitivity and specificity for Elia^®^ CTD-screen (Se: 79.36%; Sp: 90.79%) and QuantaFlash^®^ CTD-screen tests (Se: 87.23%; Sp: 87.14%) were extracted from the data of nine and four studies, respectively. Table 3 shows operational performance for each type of CTD.

Regarding Elia^®^ CTD-screen, in over six studies including only cohort studies, the prevalence of CTD on a 8109 patient sample was 10.22% (IC: 6.9–14.1). For those same studies, a sensitivity of 79.25% (IC: 74.26–83.84) and a specificity of 88.3% (IC: 83.81–92.14) were established (Table 3) (Figure 2A,B). By relating this prevalence with Elia^®^ CTD-screen diagnosis performance (LR+: 6.77; LR−: 0.23), we established a post-test probability to have a CTD of 43.5% if the test was positive, and of 2.6% if negative. Sensitivity and specificity for a cohort study using the QuantaFlash^®^ CTD-screen test were unable to be pooled, but a single cohort study was extracted.

## 4. Discussion

This meta-analysis provides a comprehensive and quantitative assessment of the diagnostic performance of CTD-screen assays in the context of systemic autoimmune rheumatic diseases. These conditions, including systemic lupus erythematosus (SLE), systemic sclerosis (SSc), SS, myositis, and undifferentiated connective tissue diseases (UCTDs), pose significant diagnostic challenges due to their overlapping clinical manifestations and variable autoantibody profiles [1,2,3]. The ability to detect disease-specific antibodies early in the disease process is critical to guide accurate diagnosis. This meta-analysis identified 11 studies (cohort studies and case–control studies) studying the performance of CTD screen to diagnose systemic rheumatic diseases [5,8,10,14,15,16,17,18,19,20,21]. This showed a sensitivity of 79.36% and a specificity of 90.79% for Elia^®^ and a sensitivity of 87.23% and a specificity of 83.56% for QuantaFlash^®^. Among cohort studies including all hospital laboratory requests, we found a prevalence of pooled CTD of 10.33%, a sensitivity of 79.25%, and a specificity of 88.3% for the Elia^®^ CTD screen. Since only one cohort study was carried out with QuantaFlash^®^ CTD screen, it was not possible to establish pooled probabilities for this kit. These findings highlight the utility of CTD screens as complementary tools in the diagnostic process but also underscore their limitations if used in isolation. Their strength lies in complementing clinical evaluation and more established serological methods. With proper application, these assays can enhance diagnostic efficiency, minimize unnecessary testing, and ultimately contribute to better patient care.

The transition of indirect immunofluorescence (IIF) from primary clinical application to a confirmatory and research role represents a significant paradigm shift in autoimmune diagnostics. As stated in our methodology, “indirect immunofluorescence (IIF) assay on cultured human epithelial carcinoma cells (Hep-2 cell) has been used as a gold standard method. However, it has been shown that some subtypes of ANA, especially anti-SSA/Ro and anti-JO 1 antibodies, may be overlooked by IIF. Furthermore, IIF requires experienced and well-trained analysts, is time-consuming, and shows high inter-observational variability”.

This evolution occurred due to several converging factors: technical limitations where IIF misses certain clinically important antibodies (anti-SSA/Ro, anti-Jo1); practical constraints including the requirement for specialized expertise, time-intensive protocols, and significant inter-observer variability; and automation advantages where CTD screens enable simultaneous detection of multiple antibodies with standardized, reproducible results suitable for high-throughput clinical laboratories.

However, this shift should not be interpreted as IIF becoming obsolete. Rather, IIF has evolved into a confirmatory and specialized diagnostic tool that provides irreplaceable pattern-specific diagnostic intelligence, while automated CTD screens have assumed the role of efficient initial screening in appropriate clinical contexts.

The practical application of these diagnostic modalities to clinical specimens requires specific technical considerations that are essential for optimal diagnostic performance. For serum-based testing, standard venipuncture with serum separation is followed by serial dilutions (typically 1:80 to 1:160 for IIF, automated dilution for CTD screens). IIF application involves serum application to HEp-2 cell substrates, incubation, washing, fluorescent secondary antibody application, and expert microscopic interpretation. CTD screen application utilizes automated liquid handling systems with standardized reagent addition, incubation protocols, and optical detection systems. For tissue-based diagnosis, while our study focuses on serum-based assays, direct immunofluorescence on tissue biopsies (particularly skin and kidney) provides complementary diagnostic information, especially valuable in lupus nephritis assessment and cutaneous manifestations of systemic autoimmune diseases. This approach detects in situ immune complex deposition and complements serum-based antibody detection. The integration of both approaches—serum-based antibody detection and tissue-based immune complex visualization—provides comprehensive diagnostic assessment, though tissue-based methods were beyond the scope of this meta-analysis.

The detection of ANA is important to help in the diagnosis of CTD, but it should be used with care. Because of its good sensitivity, ANA detection provides a good screening test, but ANAs are frequently found in healthy individuals or in patients with other diseases, with a prevalence between 5 and 30% depending upon studied populations [22]. High clinical suspicion is necessary to avoid useless and costly investigations in the case of false positive tests. In this way, a sensible approach in using an ANA test is essential, and testing should only be conducted in the appropriate clinical context, with patients’ characteristic symptoms and clinical examination ideally coming before laboratory tests. Another consideration is the potential utility of CTD screens in the early identification of patients with UCTDs or overlap syndromes. These patient subsets often evolve over time into defined autoimmune conditions, and early serological signals can be pivotal in risk stratification and longitudinal monitoring. As such, CTD screens may have prognostic as well as diagnostic value, though this hypothesis warrants further investigation in prospective cohort studies. The economic implications of ANA and CTD screen testing in apparently healthy individuals present a significant healthcare challenge. The economic burden of inappropriate screening includes unnecessary specialist referrals generating substantial healthcare costs, patient anxiety and quality-of-life impacts from false-positive results, cascade testing and follow-up investigations in low-probability patients, and healthcare system resource diversion from patients with genuine clinical need. Cost-effectiveness analysis suggests that ANA and CTD screen testing should be reserved for patients with intermediate-to-high pretest probability based on clinical presentation. The positive predictive value in low-prevalence populations renders population-based screening economically unsustainable and clinically counterproductive. High clinical suspicion is necessary to avoid useless and costly investigations in cases of false positive tests.

Screening value in asymptomatic individuals is therefore limited not only by poor positive predictive value but also by the substantial economic and psychological costs associated with false-positive results in healthy populations. A significant opportunity for improving diagnostic performance lies in integrating CTD screens with readily available hematologic parameters. Complete blood count (CBC) abnormalities are frequently present in connective tissue diseases and can enhance diagnostic accuracy when combined with serological testing. Key hematologic parameters in CTD include lymphopenia (common in SLE and other CTDs), thrombocytopenia (particularly in SLE and antiphospholipid syndrome), anemia of chronic disease (prevalent across CTDs), and elevated inflammatory markers (ESR, CRP). Calculated ratios such as platelet–lymphocyte ratio (PLR) can provide additional diagnostic value when integrated with immunological markers. The combination of CBC parameters with ELISA/CTD screen results may be mandatory for optimal diagnostic accuracy in rheumatic disorders. This integrated approach leverages the complementary information provided by cellular immune dysfunction (reflected in CBC abnormalities) and humoral immune activation (detected by antibody testing). Economic advantages of this combined approach include utilization of routinely ordered laboratory tests without additional cost, enhanced diagnostic accuracy potentially reducing the need for additional testing, and improved risk stratification for specialist referral decisions. Our analysis supports a comprehensive diagnostic strategy combining three complementary assessment domains: ANA/CTD antibody testing for humoral immune activation detection, complete blood count analysis for cellular immune dysfunction assessment, and clinical findings integration for phenotypic disease manifestation evaluation. This triple approach offers several advantages: comprehensive immune system assessment addressing both humoral and cellular components, enhanced sensitivity and specificity through multi-parameter integration, cost-effective utilization of standard laboratory tests, and improved clinical decision making through integrated data interpretation. While achieving 100% sensitivity and specificity remains unlikely given the complexity of autoimmune diseases and existence of seronegative cases, this integrated approach significantly improves diagnostic accuracy compared to single-parameter testing. The synergistic effect of combining serological, hematologic, and clinical parameters provides a more robust diagnostic framework than relying on any single testing modality. Implementation considerations include standardized protocols for parameter integration, training for multi-parameter interpretation, quality assurance across testing domains, and cost-effectiveness validation in diverse clinical settings. Our findings support a structured, tiered approach to autoimmune diagnostics that optimizes both diagnostic accuracy and resource utilization. CTD screens should serve as initial diagnostic tools specifically in patients with intermediate-to-high pretest probability for connective tissue disease, not as broad population screening instruments. This targeted approach addresses the fundamental limitation that positive predictive value remains modest in low-prevalence populations. All positive CTD screen results should be confirmed by indirect immunofluorescence (IIF) to maximize diagnostic accuracy and provide crucial pattern-specific diagnostic information. This confirmation step is essential because automated CTD screens, while efficient and standardized, cannot provide the diagnostic intelligence offered by expert pattern recognition. IIF pattern recognition provides diagnostic intelligence that cannot be replicated by solid-phase assays. Specific immunofluorescence patterns offer valuable diagnostic clues that guide both diagnosis and subsequent testing strategies:Peripheral/homogeneous patterns are strongly associated with anti-dsDNA antibodies and systemic lupus erythematosus.Centromere patterns are highly suggestive of limited systemic sclerosis.Nucleolar patterns are characteristic of systemic sclerosis, particularly diffuse cutaneous forms.Speckled patterns encompass various specificities including anti-Sm, anti-RNP, anti-SSA/Ro, and anti-SSB/La antibodies.

Accurate interpretation of these patterns requires experienced laboratory personnel and should be integrated with clinical assessment. The diagnostic accuracy demonstrated in our meta-analysis for CTD screens should therefore be viewed as complementary to, rather than a replacement for, expert IIF interpretation.

CTD screens are valuable in the initial diagnostic phase for detecting systemic autoimmune diseases. However, CTD screens are not recommended for monitoring disease progression or treatment response. For longitudinal patient management, it is essential to adopt a personalized approach based on the involved organ system and utilize appropriate disease-specific markers. This phase-specific approach optimizes both clinical outcomes and healthcare resource allocation.

The increased prevalence of non-organ-specific autoantibodies—particularly antinuclear antibodies (ANA)—has been observed in individuals with X-chromosome aneuploidies, such as Klinefelter syndrome [23]. This finding underscores that autoantibody production may be influenced by genetic factors independent of autoimmune disease, thereby complicating standard diagnostic interpretation.

A positive ANA result alone is insufficient to establish a diagnosis of systemic autoimmune disease. ANA positivity can occur in a wide range of conditions—including infections, neoplasms, drug exposures, and even among healthy individuals—with its prevalence varying according to demographic, genetic, and methodological factors.

Based on our findings and the need for practical implementation guidance, we propose the following integrated diagnostic algorithm:Clinical Assessment: Comprehensive evaluation including symptoms, physical examination, and organ system involvement.Pretest Probability Determination: Low probability—avoid testing; intermediate–high probability—proceed with a multi-parameter approach.Multi-Parameter Testing: CTD screen + CBC with calculated ratios + inflammatory markers.Result Integration: Combined interpretation of serological and hematologic findings.Confirmation Strategy: Positive CTD screen confirmed with IIF and expert pattern interpretation.Clinical Correlation: Rheumatological integration of all findings with clinical presentation.Monitoring Framework: Disease-specific, organ-based follow-up rather than repeat CTD screening.

The QUADAS-2 assessment showed that study quality was generally moderate, with most concerns focused on patient selection and reference standard application. In this context, clinicians should be cautious in generalizing the performance of these tests to all patient populations and should consider the risk of spectrum bias, particularly in settings outside of tertiary care or specialist centers. Despite these limitations, the diagnostic accuracy of the Elia^®^ and QuantaFlash^®^ CTD screens remains robust, particularly when used judiciously. Another strength of our study is the inclusion of both cohort and case–control studies across a wide geographic and clinical range. This diversity strengthens the generalizability of our findings but also introduces heterogeneity. While pooled estimates provide meaningful summary statistics, the individual study designs and selection criteria varied, particularly regarding the definition of control populations and reference standards for final diagnosis. In some studies, diagnoses were only confirmed in patients with positive CTD screens or IIF, introducing potential verification bias. Moreover, a few studies included healthy blood donors as controls, which may overestimate specificity due to the lack of clinical ambiguity in this population.

An important implication of this work is the necessity to interpret CTD screen results within the framework of pretest probability. In clinical settings with low disease prevalence, such as primary care or general internal medicine, the positive predictive value of ANA and CTD screen testing is modest. This is consistent with previous studies demonstrating high ANA positivity rates in healthy individuals, ranging from 5% to 30%, depending on demographic and methodological variables. As a result, indiscriminate testing can lead to overdiagnosis, unnecessary specialist referrals, and unwarranted patient anxiety. Conversely, in rheumatology clinics or in patients with suggestive clinical features, these tests can meaningfully contribute to the diagnostic process when integrated into a structured clinical assessment.

Our results indicate that the Elia^®^ CTD-screen had a pooled sensitivity of 79.36% and specificity of 90.79%, while the QuantaFlash^®^ assay demonstrated slightly higher sensitivity (87.23%) with slightly lower specificity (83.56%). These findings suggest that both assays can contribute to the detection of systemic rheumatic diseases, but their use must be contextualized. Sensitivity and specificity varied among different CTDs. For example, primary Sjögren’s syndrome and mixed connective tissue disease (Sharp’s syndrome) showed the highest sensitivities, while autoimmune myositis exhibited relatively lower sensitivity. These results reflect the underlying differences in antibody production and the composition of each test panel and suggest that no single test can reliably identify all CTDs with equal accuracy.

According to Abeles et al., 90% of referred patients to the Rheumatology Department had no evidence of rheumatic disease associated to ANAs identification [7], which means that the prevalence of CTD is only of 10% and so the majority of people with positive ANA testing are not affected by a systemic rheumatic disease. Moreover, ANA positivity has a low positive predictive value, which can be explained by using tests with suboptimal performance in groups with low probability clinical presentation [14].

The term ‘CTD screen’ may be misleading and overly promotional, which can be confusing because we cannot really consider it as a screening test or use it as a diagnostic test. CTD-screen may be interesting to help CTD diagnosis, but clinician expertise remains, essential especially because some CTDs are seronegative.

## 5. Conclusions

ANAs are a cornerstone in detection of connective tissue disease, but they are widely spread in the healthy population too. CTD-screen does not have diagnostic performance if used alone in clinical practice for the detection of CTD. But this test could help specialists to make diagnosis or perform other complementary exams. Currently, CTD diagnosis is complex, and specialist assessment remains essential. The optimal diagnostic approach integrates multiple complementary modalities: automated CTD screens for efficient initial detection, expert IIF interpretation for pattern-specific diagnostic intelligence, hematologic parameter analysis for cellular immune assessment, and comprehensive clinical evaluation for phenotypic disease characterization. This multi-parameter strategy maximizes diagnostic accuracy while optimizing healthcare resource utilization through targeted testing, appropriate confirmation protocols, and recognition of the distinct roles of different diagnostic modalities in the evolving landscape of autoimmune diagnostics. The findings of this meta-analysis support a paradigm shift toward integrated, multi-parameter autoimmune diagnostics that combines the efficiency of modern automated systems with the irreplaceable value of expert interpretation and comprehensive clinical assessment. This approach acknowledges both the technical evolution of diagnostic capabilities and the economic imperatives of sustainable healthcare delivery while maintaining the diagnostic accuracy essential for optimal patient care. Future research should explore the integration of clinical prediction tools with serological testing to optimize diagnostic algorithms. Machine learning and artificial intelligence could assist in combining clinical variables, imaging, and serology to improve diagnostic precision and reduce false-positive rates. Additionally, cost-effectiveness studies are needed to evaluate the long-term impact of CTD screen utilization on healthcare resource allocation and patient outcomes.

## Figures and Tables

**Figure 1 antibodies-14-00056-f001:**
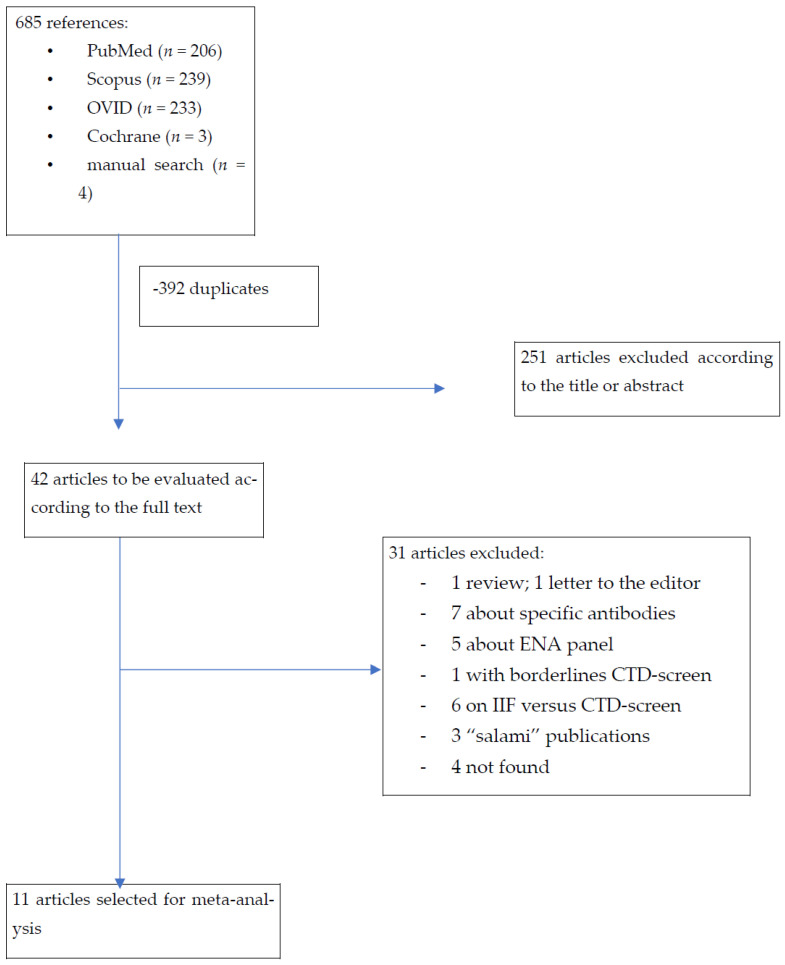
Flowchart of the selection of the different studies used in this meta-analysis.

**Figure 2 antibodies-14-00056-f002:**
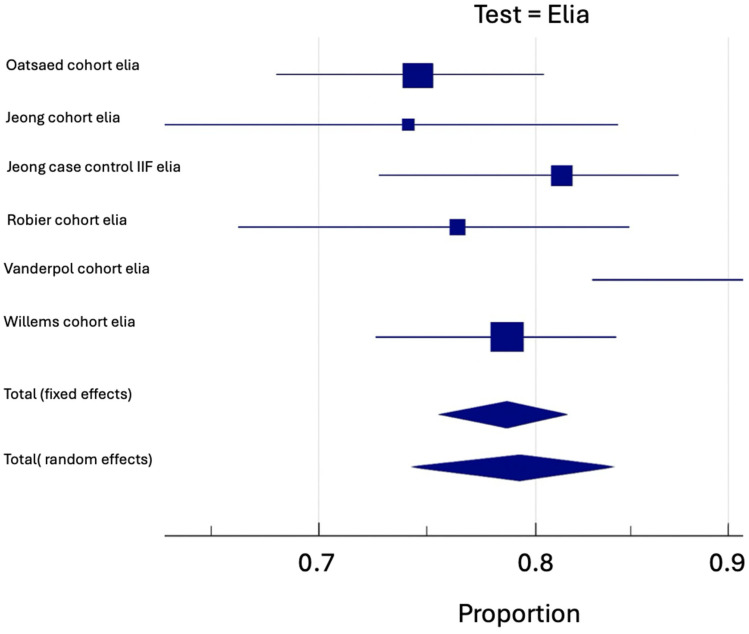
(**A**) Funnel and Forest plot for cohort sensitivity using the Elia^®^ CTD-screen test. (**B**) Funnel and Forest plot for cohort specificity using the Elia^®^ CTD-screen test.

**Table 1 antibodies-14-00056-t001:** Results from the studies included in the meta-analysis.

	All Connective Tissue Diseases		Systemic Lupus Erythematosus		Primary Sjogren’s Syndrome		Systemic Scleroderma		Dermato-Polymyositis		Undifferentiated Connective Tissue Diseases		Controls Without Connective Tissue Diseases		Blood Donor	
Author	CTDpos	total	CTDpos	total	CTDpos	total	CTDpos	total	CTD pos	total	CTD pos	total	CTD neg	total	CTD neg	total
Bizarro Cohort study Elia-QuantaFlash	282	368	91	123	111	128	47	66	25	39	8	12				
Willems Cohort study Elia	170	216	62	83	43	45	54	63	12	17	8	8	1804	2197		
Robier Cohort study Elia	65	85	21	28	17	17	10	11		1	4	4	1523	1623		
Van der Pool Cohort study Elia	66	72	43	44	13	16	2	4	4	4	4	4	210	250		
Van der Pool Cohort study QuantaFlash	71	72	44	44	15	16	4	4	4	4	4	4	190	250		
Van der Pool Case–control study Elia	104	120	36	40	33	34	17	23	18	23						
Van der Pool Case–control study QuantaFlash	101	120	38	40	32	34	17	23	14	23						
Lopez-Hoyos Case–control study Varelisa	193	254	152	202	30	41	11	11					198	218	95	105
Bentow Case–control study QuantaFlash	139	178	79	98	24	30	21	30			15	20	192	204	140	146
Jeong Cohort study Elia	46	62	31	35		2		2			13	23	898	1031		
Jeong Cohort study Elia	91	112	57	67	16	19	13	21			5	5	924	1003		
Claessens Case–control study Elia	386	480	95	119	59	65	181	220	25	50	26	26	748	767	276	279
Claessens Case–control study QuantaFlash	412	480	102	119	59	65	192	220	33	50	26	26	675	767	262	279
Op De Beeck Case–control study Elia	171	236	59	80	32	36	50	69	11	28	13	13	409	422	145	149
Olsaed Cohort study Elia	150	201	?	142	?	24	?	15	?	10			1112	1257		

? unknown

**Table 2 antibodies-14-00056-t002:** Quality assessment of the studies according to the QUADAS-2 score tool.

Author and Test	Year	Country And Study	Quality Assessment of the Studies (QUADAS2)
Risk of Bias	Applicability Concerns
Patients’ Selection	Index Test	Standard Reference	Flow and Timing	Patients’ Selection	Index Test	Standard Reference
Bizzaro Elia-Quanta	2018	Italy Cohort study		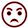	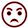	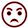			
Willems Elia	2018	Belgium Cohort study							
Robier Elia	2016	Austria Cohort study							
Van der Pool Elia	2018	Netherlands Cohort study							
Van der Pool Quanta	2018	Netherlands Cohort study							
Van der Pool Elia	2018	Netherlands Case–control study	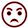				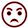		
Van der Pool Quanta	2018	Netherlands Case–control study	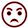				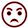		
Lopez-Hoyos Varelisa	2007	Spain Case–control study	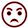				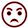		
Bentow Quanta	2015	International Case–control study	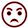				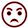		
Jeong Elia	2018	Korea Cohort study	?				?		
Jeong Elia	2017	Korea Cohort study	?				?		
Claessens Elia	2018	International Case–control study	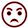				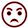		
Claessens Quanta	2018	International Case–control study	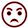				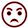		
Op De Beeck Elia	2011	Belgium Case–control study	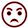				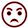		
Alsaed Elia	2018	Qatar Cohort							

*Legend*: 
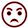
 High 

 low ? unclear.

**Table 3 antibodies-14-00056-t003:** Operational performance for each type of CTD.

**ELIA© CTD-SCREEN**								
**Sensitivity for…**	**Number of Studies**	**Number of Cases**	**Pooled Sensitivity**	**IC95%**	**Q**	**I^2^**	**Egger**	**Begg**
Connective tissue disease diagnosis	9	1584	79.36	75.61–82.88	23.54	66.02%	NS	NS
Lupus erythematosus diagnosis	8	496	82.98	76.49–88.60	22.52	68.92%	NS	NS
Primary Sjogren’s syndrome diagnosis	7	232	91.43	86.69–95.21	8.2356	27.15%	NS	NS
Systemic scleroderma diagnosis	7	411	77.44	70.40–87.78	11.21	46.46%	NS	NS
Autoimmune myositis diagnosis	6	123	60.95	43.13–77.37	16.624	69.92	NS	NS
Sharp’s syndrome diagnosis	7	83	93.05	77.92–99.80	24.13	75.14	NS	NS
**Specificity for …**								
Clinic controls (no connective tissue disease)	7	8550	91.05	86.59–94.69	298.65	97.66	NS	NS
Healthy controls (blood donor)	2	428	98.18	96.09–99.49	1.7005	41.19	*p* <0.001	NS
Total pooled controls	8	8122	90.79	86.69–94.20	233.28	97	NS	NS
**ELIA© Pooled Cohorts**	**Number of Studies**	**Number of Cases**	**Pooled %**	**IC95%**	**Q**	**I^2^**	**Egger**	**Begg**
Prevalence of connective tissue disease	6	8109	10.22	6.90–14.10	141.63	96.47	NS	NS
Sensitivity	6	748	79.25	74.26–83.84	12.64	60.43	NS	NS
Specificity	6	7361	88.3	83.81–92.14	151.71	96.7	NS	NS
**QUANTAFLASH© CTD-SCREEN**								
**Sensitivity for…**	**Number of Studies**	**Number of Cases**	**Pooled Sensitivity**	**IC95%**	**Q**	**I^2^**	**Egger**	**Begg**
Connective tissue disease diagnosis	4	850	87.23	79.10–93.58	25.28	88.13	NS	NS
Lupus erythematosus diagnosis	4	301	91.13	80.51–97.83	20.94	85.68	NS	NS
Primary Sjogren’s syndrome diagnosis	4	145	89.1	83.31–93.79	3.2471	7.61	NS	NS
Systemic scleroderma diagnosis	4	277	80.52	68.07–90.48	7.892	61.93%	NS	NS
Autoimmune myositis diagnosis	3	77	68.205	52.30–82.18	3.36	40.62%	NS	NS
Sharp’s syndrome diagnosis	3	50	91.65	67.02–99.94	9.21	78.28	NS	NS
**Specificity for …**								
Clinic controls (no connective tissue disease)	3	796	83.56	75.18–90.50	13.3791	85.05	NS	NS
Healthy controls (blood donor)	2	425	94.474	92.11–96.44	0.5871	0	*p* <0.001	NS
Total pooled controls	3	1221	87.14	77.18–94.56	35.15	94.31	NS	NS

NS: Not significant.

## Data Availability

No new data was created in this study.

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
