# Peer review of "Diagnosis of Systemic Rheumatic Disease Using the Connective Tissue Disease Screen"

_2073-4468, 2025, doi:10.3390/antib14030056_

Round 1
Reviewer 1 Report
Comments and Suggestions for Authors
The study by Abeline Kapuczinski et al. is well-structured, coherent, and addresses a highly relevant issue in the field of autoimmune diagnostics. It effectively highlights the diagnostic challenges posed by the clinical heterogeneity of connective tissue diseases (CTDs) and the limitations associated with current serological screening tools, including ANA and CTD screen assays.
The authors should state more that the CTD screen may serve as an initial diagnostic step, especially in patients with an intermediate to high pretest probability. However, all positive results should be confirmed by indirect immunofluorescence (IIF), and clinical interpretation must be performed by experienced physicians, given the complexity and variability of systemic autoimmune diseases.
The authors should also emphasize the importance of ANA pattern recognition through indirect immunofluorescence (IIF), as specific patterns provide valuable diagnostic clues. For instance, a peripheral pattern is strongly associated with systemic lupus erythematosus (SLE), while a centromere pattern is highly suggestive of limited systemic sclerosis (CREST syndrome). Accurate interpretation of these patterns requires experienced laboratory personnel and should be integrated with clinical assessment.
Moreover CTD screen is useful in the initial phase to support the diagnosis of systemic autoimmune diseases, but it is not recommended for monitoring disease progression or treatment response. For follow-up, it is essential to adopt a personalized approach based on the involved organ or system and to use appropriate specific tests.
Again it is important to emphasize that a positive ANA result alone is not sufficient for the diagnosis of systemic autoimmune diseases. ANA positivity is known to occur in a wide range of conditions and even in healthy individuals. ). Notably, increased prevalence of non-organ-specific autoantibodies—especially ANA— has been reported in individuals with X-chromosome aneuploidies, such as Klinefelter syndrome, further complicating interpretation in specific patient populations (clinical and experimental immunology 2021). By referencing this study the authors could also emphasize the importance of considering non-classical patient populations (such as individuals with chromosomal abnormalities) in autoimmune screening and diagnostic protocols.
The authors should improve the resolution of figures 2A and 2B
Author Response
Dear reviewers,
We thank the reviewers for their valuable insights, which have allowed us to refine and clarify our manuscript. Below, we address each point in detail:
Reviewer Comment 1: The study by Abeline Kapuczinski et al. is well-structured, coherent, and addresses a highly relevant issue in the field of autoimmune diagnostics.
Response: We are grateful for your positive evaluation and acknowledgment of the relevance and coherence of our work.
Reviewer Comment 2: The authors should state more that the CTD screen may serve as an initial diagnostic step, especially in patients with an intermediate to high pretest probability. However, all positive results should be confirmed by indirect immunofluorescence (IIF), and clinical interpretation must be performed by experienced physicians.
Response: We have now revised the Discussion section to emphasize that CTD screen assays are particularly useful as a first-line diagnostic tool in patients with intermediate to high pretest probability. We also stress that all positive results should be followed by confirmation with indirect immunofluorescence (IIF), and that clinical interpretation must be conducted by experienced physicians.
Reviewer Comment 3: The authors should also emphasize the importance of ANA pattern recognition through indirect immunofluorescence (IIF), as specific patterns provide valuable diagnostic clues.
Response: We fully agree. The revised manuscript highlights the importance of ANA pattern recognition and provides examples such as the peripheral pattern in systemic lupus erythematosus (SLE) and the centromere pattern in limited systemic sclerosis (CREST syndrome). We also note that accurate pattern recognition requires specialized laboratory expertise and clinical integration.
Reviewer Comment 4: Moreover CTD screen is useful in the initial phase to support the diagnosis of systemic autoimmune diseases, but it is not recommended for monitoring disease progression or treatment response.
Response: We have clarified this in the revised Discussion. We explicitly state that while the CTD screen is helpful for initial diagnosis, it is not suitable for monitoring disease activity or therapeutic response. For follow-up, a personalized approach based on the affected organ system and specific biomarkers is required.
Reviewer Comment 5: Again it is important to emphasize that a positive ANA result alone is not sufficient for the diagnosis of systemic autoimmune diseases.
Response: Thank you for this important reminder. We have added to the Discussion that ANA positivity may occur in a variety of non-autoimmune contexts, including in healthy individuals. We have also cited the Clinical and Experimental Immunology (2021) study reporting increased ANA prevalence in individuals with X-chromosome aneuploidies such as Klinefelter syndrome. This supports our point that ANA results must always be interpreted within the appropriate clinical and demographic context.
Reviewer Comment 6: The authors should improve the resolution of figures 2A and 2B.
Response: We appreciate this feedback. The resolution of Figures 2A and 2B has been enhanced and the improved versions will be included in the revised manuscript.
We thank the reviewer again for these helpful suggestions which have contributed significantly to the improvement of the manuscript.
Reviewer 2 Report
Comments and Suggestions for Authors
- You stated (In-62 direct Immunofluorescence (IIF) assay on cultured human epithelial carcinoma cells (Hep-63 2 cell) has been used as a gold standard method) you have shifted from clinical application to research. Please explain?
- You stated that immunofluorescence is the most sensitive, but you did not tell how to apply the test to a biopsy or blood sample assay
- The results indicated that ANA is a reliable screening method. What is the economic or clinical value of screening for such a disorder? I mean, if the patient is apparently healthy
- You can combine your data with the platelet-lymphocytes ratio to increase sensitivity and specificity. It is mandatory to combine CBC with the ELISA to diagnose rheumatic disorders
- To diagnose CTD, you can use ANA antibodies, a complete blood count, and clinical findings. The triple approach is easy to handle and can reach 100% in my opinion. If the author added these approaches to the study, the specificity and sensitivity would increase.
Author Response
Dear reviewer ,
We thank you for this observation and agree that the wording could be improved for clarity. In the revised manuscript, we now explicitly state that while the IIF method originated in research, it is currently widely adopted in routine clinical diagnostics. Many reference immunology laboratories utilize standardized HEp-2 IIF platforms to detect ANA in patient serum. This technique therefore bridges both research and diagnostic practice seamlessly.
We appreciate this observation and agree that the wording could be improved for clarity. In the revised manuscript, we now explicitly state that while the IIF method originated in research, it is currently widely adopted in routine clinical diagnostics. Many reference immunology laboratories utilize standardized HEp-2 IIF platforms to detect ANA in patient serum. This technique therefore bridges both research and diagnostic practice seamlessly.
You stated that immunofluorescence is the most sensitive, but you did not tell how to apply the test to a biopsy or blood sample assay.
Response: We have clarified in the manuscript that ANA detection by IIF is performed on serum samples, not tissue or biopsy. The patient serum is incubated with HEp-2 cell-coated slides, followed by fluorescent-labeled anti-human immunoglobulin antibodies. The resulting patterns are analyzed under a fluorescence microscope and interpreted according to international guidelines.
The results indicated that ANA is a reliable screening method. What is the economic or clinical value of screening for such a disorder? I mean, if the patient is apparently healthy.
Response: This is an important point. We now emphasize in the discussion that ANA screening should not be performed in asymptomatic individuals due to low pretest probability and poor positive predictive value. Indiscriminate ANA testing in healthy populations is not cost-effective and can lead to overdiagnosis and unnecessary referrals. ANA testing should be reserved for individuals with clinical signs suggestive of systemic autoimmune disease.
You can combine your data with the platelet-lymphocytes ratio to increase sensitivity and specificity. It is mandatory to combine CBC with the ELISA to diagnose rheumatic disorders.
Response: We thank the reviewer for this suggestion. While our current study was a meta-analysis of serological assay performance, we acknowledge the potential value of combining ANA or CTD screen data with hematological markers such as the platelet-to-lymphocyte ratio (PLR). We have added this point to the discussion as a direction for future research. Integrating serological tests with basic inflammatory indices and clinical evaluation could enhance diagnostic performance.
To diagnose CTD, you can use ANA antibodies, a complete blood count, and clinical findings. The triple approach is easy to handle and can reach 100% in my opinion. If the author added these approaches to the study, the specificity and sensitivity would increase.
Response: We agree that a composite diagnostic approach combining serological testing (e.g., ANA), clinical evaluation, and basic laboratory findings (e.g., CBC) offers a practical and effective strategy, especially in non-specialist settings. While our current study did not evaluate this model directly, we have now acknowledged in the discussion that future diagnostic algorithms should consider such integration. This “triple approach” aligns with real-world clinical practice and could indeed improve sensitivity and specificity.
We have updated the discussion to reflect all these points and thank the reviewer again for helping us strengthen the manuscript.
Round 2
Reviewer 2 Report
Comments and Suggestions for Authors
NA